# Melatonin Modulates ZAP70 and CD40 Transcripts via Histone Modifications in Canine Ileum Epithelial Cells

**DOI:** 10.3390/vetsci12020087

**Published:** 2025-01-23

**Authors:** Jian Hong, Saber Y. Adam, Shiqi Wang, Hao Huang, In Ho Kim, Abdelkareem A. Ahmed, Hao-Yu Liu, Demin Cai

**Affiliations:** 1School of Marine and Biological Engineering, Yancheng Teachers University, Yancheng 224002, China; hongj01@yctu.edu.cn; 2College of Animal Science and Technology, Yangzhou University, Yangzhou 225009, China; mh23108@stu.yzu.edu.cn (S.Y.A.); 221902318@stu.yzu.edu.cn (S.W.); 221902107@stu.yzu.edu.cn (H.H.); 007725@yzu.edu.cn (H.-Y.L.); 3Jiangsu Key Laboratory of Animal Genetic Breeding and Molecular Design, College of Animal Science and Technology, Yangzhou University, Yangzhou 225009, China; 4Department of Animal Resource and Science, Dankook University, Cheonan 100000, Republic of Korea; inhokim@dankook.ac.kr; 5Department of Veterinary Sciences, Botswana University of Agriculture and Natural Resources, Gaborone P.O. Box 100, Botswana; aabdallah@buan.ac.bw; 6Biomedical Research Institute, Darfur University College, Nyala P.O. Box 160, Sudan; 7Department of Physiology and Biochemistry, Faculty of Veterinary Science, University of Nyala, Nyala P.O. Box 155, Sudan

**Keywords:** melatonin, transcriptome, ileum, canine, histone modifications

## Abstract

Melatonin (MLT), a hormone produced by the pineal gland, plays a crucial role in regulating inflammation in mammals. Our study investigates how melatonin influences gene expression in canine ileum epithelial cells, focusing on the modulation of inflammatory genes ZAP70 and CD40 through histone modifications. We found that MLT treatment upregulates key immune response genes and alters the NF-kappa B signaling pathway, offering potential therapeutic insights for inflammatory bowel diseases in dogs. This research provides a clear understanding of MLT’s anti-inflammatory effects, highlighting its value for pet health and veterinary medicine.

## 1. Introduction

Melatonin (N-acetyl-5-methoxytryptophan, MLT), a neurohormone produced by the pineal gland in vertebrates, has a circadian rhythm. The hypothalamic suprachiasmatic nucleus regulates the circadian rhythm of MLT secretion, leading it to secrete at a higher level at night [1]. Beyond the pineal gland, MLT is also produced by the retina and, under certain conditions, by the skin and gastrointestinal system [2]. As a darkness signal, it synchronizes endocrine cycles by communicating with the brain and other organs. This modulation affects the timing elements of both central and peripheral biological clocks. MLT is widely recommended for mitigating jet lag, a sleep rhythm disruption caused by transoceanic flights crossing multiple time zones [3,4]. Its therapeutic potential is supported by studies highlighting various effects, including analgesic, immune-boosting, anticancer, antioxidative, and cytotoxic properties [5]. According to some studies, MLT therapy increases the weight of immune organs in healthy and immunocompromised individuals [6,7]. Moreover, MLT anti-proliferative effects have been shown in vitro in human lymphocytes stimulated by PHA [8]. MLT also regulates the proliferation of immunocompetent cells, which affects both the innate and specific immune responses [9] and secretion of immune mediators, like cytokines [10]. Furthermore, studies have shown that MLT plays a regulatory role in apoptosis, autophagy, and angiogenesis during the development of colorectal cancer [11]. MLT has many biological effects, making it crucial for controlling how genes work, particularly those related to epigenetics [12]. In recent years, its role has grown to include a substantial influence on the immune system, highlighting its ability to alter the gene expression in immune cells. Furthermore, the protection of canine intestinal epithelial cells and the regulation of immune function may also play a beneficial role in the subsequent prevention of obesity in dogs [13].

T cell receptor (TCR) signaling is facilitated by two nonreceptor tyrosine kinases: zeta-chain-associated protein kinase ZAP-70 and Src family tyrosine kinase Lck [14]. ZAP-70 is a tyrosine kinase mainly expressed in T cells, a subdivision of B and NK cells [15]. Tandem SH2 domains in its molecular structure connect to phosphorylated immunotyrosine-based activation motifs (ITAMs), which are selectively activated by phosphorylating tyrosines 315, 319, and 493 [16,17]. ZAP-70′s activation loop is stabilized by the phosphorylation, which also encourages the phosphorylation of other proteins like SLP-76 (Src homology two domain-containing leukocyte phosphoprotein of 76 kDa) and LAT (linker for the activation of T cells), promoting more downstream signaling [18], leading to the recruitment of effector proteins that stimulate T cell activation [19]. The loss of ZAP-70 function causes severe combined immunodeficiency (SCID). This condition is characterized by a lack of functional peripheral T cells in both humans and animals. This highlights the importance of ZAP-70 for T cell development and activation [20]. The CD40 antigen belongs to the superfamily of tumor necrosis factor receptors (TNFRs), which is expressed on many types of normal cells, such as monocytes, dendritic cells, and B cells [21], as well as on transformed cells. It is well known that CD40 and CD40 ligands (CD40L) are stimulatory immune checkpoints. They are involved in various immunological processes contributing to humoral and cell-mediated immune responses [22]. Furthermore, during active inflammatory bowel disease (IBD), intestinal epithelial cells (IECs) express CD40, indicating that CD40 has a role in stimulating proinflammatory responses [23]. The inflammatory environment in the ileum is further exacerbated by the interaction between CD40 and its ligand, triggering the release of inflammatory cytokines like IL-8 [24]. Together, these transcripts highlight the significance of immune-epithelial interactions for canine gastrointestinal health and provide possible therapeutic targets for inflammation disorders.

Histone modifications, such as methylation, acetylation, phosphorylation, and ubiquitylation [25], are pivotal in regulating gene transcription and cellular responses to environmental changes. Histone H3 acetylation on lys9 and lys27 (H3K9ac and H3K27ac) is an acetylation example that typically correlates with active transcription [26]. In contrast to acetylation, the degree and site of methylation can affect the transcriptional activity of modifications on lysine (Lys) residues. Tri-methylation on lys4 (H3K4me3) has been associated with transcriptional activation, whereas tri-methylation on histone H3 on lys9 and lys27 (H3K9me3 and H3K27me3) indicates transcriptional suppression [27]. According to studies, specific genes, such as Zip14, can be ablated to reduce histone deacetylase (HDAC) activity, raising histone acetylation and increasing the expression of genes linked to inflammation [28]. This shows that histone alterations can influence the expression of ZAP70 and CD40 by changing the chromatin landscape and promoting increased transcription factor (TF) occupancy and transcriptional activation. In this sense, it has been demonstrated that MLT affects histone modifications, which are essential for controlling gene expression in several diseases [29]. Despite the increasing studies on the effects of MLT in various species, there is still a notable gap in understanding its role in canine intestinal epithelial cells and how it may influence critical immune transcript expression. However, we hypothesize that MLT influences the immunological function of canine intestinal epithelial cells by enhancing the production of these transcripts via certain histone modifications. Therefore, this research aims to clarify the mechanisms by which MLT modulates the gene expression of inflammatory pathways via histone modifications in the canine ileum epithelial cells.

## 2. Materials and Methods

### 2.1. Cell Culture and Treatment

Canine small intestinal epithelial cells were cultured in a high glucose (25 mmol/L) DMEM (cytiva, Shanghai, China) medium containing 5% fetal bovine serum (Hyclone, Logan, UT, USA), 10 ng/mL EGF (Tongli Haiyuan, Beijing, China), 5 μg/mL Insulin (Beyotime, Shanghai, China), 20 mM HEPES (Beyotime, Shanghai, China), and 1% penicillin/streptomycin (Solarbio, Beijing, China), at 37 °C with 5% CO_2_. Cells were treated under three different conditions: (1) Control group (CON), (2) MLT (5 μM dissolved in DMSO), and (3) MLT (10 μM dissolved in DMSO) (Bidde, Shanghai, China) treated for 48 h.

### 2.2. Cell Number and Cell Death Detections

Viable cell numbers were counted under a microscope with a hemocytometer chamber at 0, 24, 48, 72, and 96 h. The caspase 3/7 activity assay kit uses caspase 3/7 to catalyze the substrate Ac-DEVD-pNA, producing yellow pNA (p-Nitroaniline). The pNA has a strong absorption near 405 nm; the activity of caspase 3/7 is calculated by determining the absorbance at 405 nm. Both groups were tested using the caspase 3/7 activity assay kit.

### 2.3. RNA Sequencing and qRT-PCR

cIECs (1.5 × 10^5^ cells/well) were seeded in 6-well plates and cultured for 48 h. Following two treatments (CON and 10 μM MLT) for 48 h, the cells were washed twice with PBS and then harvested. RNA extraction was performed by adding 1 mL of TRIzol (Invitrogen, Waltham, MA, USA) according to the manufacturer’s instructions and stored at −80 °C. Subsequently, the RNA was reverse-transcribed into cDNA according to the instructions (Vazyme, Nanjing, China). The mRNA expression was determined according to the instructions (Vazyme, Nanjing, China), and its relative expression was calculated using the 2^−ΔΔCT^ method.

### 2.4. ChIP-qPCR

cIECs were treated with a 1% formaldehyde solution and incubated on a shaker for 12 min, followed by incubation with glycine for 10 min. After the supernatant was discarded, the cells were washed twice with PBS. Next, 3 mL of PBS was added to the culture dish, and the cells on the dish were scraped off using a cell brush. The cell suspension was then centrifuged at 2000 rpm for 5 min at 4 °C. After the supernatant was discarded, the cells were resuspended in lysis buffer (1 mmol/L ethylenediamine tetra acetic acid (EDTA), 50 mmol/L N-(2-hydroxyethyl) piperazine-N-ethanesulfonic acid (HEPES) pH 8.0, 0.5% NP-40, 140 mmol/L NaCl, 0.25% Triton X-100, 10% glycerol). Following another round of centrifugation at 2000 rpm for 5 min at 4 °C, the supernatant was discarded. Then, the cells were resuspended in wash buffer (1 mmol/L EDTA, 0.5 mmol/L ethyleneglycol- bis (β-aminoethyl ether)-N, N, N, N′-tetra acetic acid (EGTA), 10 mmol/L Tris pH 8.0, 200 mmol/L NaCl) and then again centrifuged at 2000 rpm for 5 min at 4 °C. The supernatant was discarded, and the cells were resuspended in a shearing buffer (0.1% sodium dodecyl sulfate (SDS), 10 mmol/L Tris HCl pH 8.0, 1 mmol/L EDTA pH 8.0). Then, the cells were sonicated and centrifuged at 12,000 rpm for 10 min. The supernatant was incubated with magnetic beads coupled with antibodies for H3K9ac, H3K18ac, H3K27ac, H3K4me1, H3K4me3, H3K9bhb, H3K18bhb, RNAPII, and RNAPII-S5P. The immune complexes were washed with LiCl wash buffer (500 mmol/L LiCl, 1% NP-40, 0.5% sodium deoxycholate, 100 mmol/L Tris pH 7.5). Proteinase K and RNase A were added for DNA extraction for the ChIP-qPCR assays.

### 2.5. Gene Enrichment Analysis

The gene set enrichment analysis (GSEA 4.1.0) software was used to identify the enriched pathway profiles. In addition, statistically enriched biological processes or pathways in differentially expressed genes (DEGs) of the GO and KEGG pathways were ranked and categorized through the Metascape database (http://metascape.org/, accessed on 7 July 2024) and DAVID (https://david.ncifcrf.gov/, accessed on 7 July 2024). GSEA enrichment analysis plots, KEGG enrichment bubble plots, cnet plots, volcano plots, and GO-pathway enrichment result circle plots were plotted through the online platform for data analysis and visualization (http://www.bioinformatics.com.cn, accessed on 14 July 2024). Meanwhile, the STRING online web platform performed a correlation analysis of differential genes for functional protein interactions (https://cn.string-db.org/, accessed on 15 July 2024).

### 2.6. Statistical Analysis

Two-tailed two groups were compared using Student’s *t*-test, and all groups were compared using variance (ANOVA) and Tukey’s post hoc test. Statistical significance was considered at *p* < 0.05. For small sample size, non-parametric statistical methods were employed to assess the significance of differences between groups. Each data set was shown as mean ± SEM. For processing, GraphPad Prism 9.0 was used.

## 3. Results

### 3.1. MLT and Cell Growth, Survival, and KEGG Enrichment in cIECs

Cell viability assessment was conducted on cIECs treated with a 10 μM MLT extract to evaluate its potential protective effects. The cell counting data revealed no significant differences in cell numbers between MLT-treated groups and CON at various time points (0, 24, 48, 72, and 96 h), as illustrated in Figure 1A. Additionally, the activity of caspase 3/7, a key indicator of apoptosis, remained unaffected by MLT treatment at concentrations of 5 μM and 10 μM, as depicted in Figure 1B. Transcriptome analysis was performed on cIECs with and without 10 μM MLT. The heatmap and volcano plot in Figure 1C,D display the differential gene expression profiles, highlighting the genes that were significantly modulated by MLT treatment compared to the CON. Gene Set Enrichment Analysis (GSEA) was employed to identify the most significantly altered pathways, as indicated by their enrichment scores (ES), which are graphically represented in Figure 1E. Notably, no substantial alterations were observed in cell cycle or proliferation-related genes in response to MLT treatment, as shown in Figure 1F. Further functional annotation of the differentially expressed transcripts is presented in Figure 2A–C. Kyoto Encyclopedia of Genes and Genomes (KEGG) pathway enrichment analysis revealed that the most significantly enriched pathways were primary immunodeficiency and the NF-kappa B (NF-κB) signaling cascade. The Protein–protein Interaction (PPI) network diagram in Figure 2D provides a visual representation of the interconnected pathways and genes that were significantly altered, offering insights into the complex regulatory networks affected by MLT. The network map in Figure 2E corroborates these findings, illustrating the intricate interplay between the enriched pathways and various cellular processes, underscoring the multifaceted nature of MLT’s influence on cIECs.

### 3.2. MLT Regulates Gene Expression in Inflammation-Related Pathways

In our study, we delved into the potential interactions among differentially expressed genes (DEGs) implicated in inflammatory and immune response pathways. Our focus was on the modulation of these pathways in response to T cell activation, with a particular emphasis on the enrichment of pathways such as NF-κB and primary immunodeficiency signaling, as well as ubiquitin-mediated proteolysis. These findings suggest the activation of stress-responsive signaling cascades and cellular mechanisms that maintain proteostasis. In the context of our research, we identified overlapping genes within these enriched pathways for both up- and downregulated DEGs, as depicted in Figure 3A,B. Our analysis of gene components centered on specific pathways revealed significant associations with KEGG pathway enrichment for genes involved in primary immunodeficiency and NF-κB signaling. Specifically, we observed an increase in the expression of CD40, ZAP70, and IL7R and a decrease in LCK, RPL37, TNFRSF13B, CD4, CD40LG, BLNK, and CIITA in the MLT-treated group relative to the CON within the primary immunodeficiency pathway, as illustrated in Figure 3E. Similarly, within the NF-κB signaling pathway, MLT treatment led to an upregulation of CD40, ZAP70, TICAM1, VCAMI, GADD45B, IRAK1, TRADD, RELA, RIPK1, and RELB, and downregulation of PRKCB, LY96, CD40LG, ILIB, BLNK, and TNFRSF11A compared to the CON, as shown in Figure 3F. These results underscore the intricate interplay between MLT treatment and the modulation of key genes within critical immune response pathways, highlighting MLT’s potential therapeutic implications in immune and inflammatory regulation.

### 3.3. Histone Modifications Facilitate the Transcriptional Suppression of ZAP70 and CD40

As shown in Figure 4A, the demethylation pathway was enriched in the MLT-treated cells using GSEA. Heat map of core gene expression of demethylation enzymes (Figure 4B). Because of epigenetic controls on primary immunodeficiency and NF-κB signaling pathway modulation, we used ChIP-qPCR to detect histone mark enrichments in the CD40 and ZAP70 genes. The histone markers like H3K9ac, H3K18ac, H3K27ac, H3K4me1, H3K4me3, H3K9bhb, H3K18bhb, RNAPII and RNAPII-S5P antibodies were measured. MLT treatments significantly increased the histone markers associated with transcriptional activation, H3K9ac (Figure 4C), H3K18ac (Figure 4D), H3K27ac (Figure 4E), H3K4me1, and H3K4me3 (Figure 4F–G), but not H3K9bhb (Figure 4H) and H3K18bhb (Figure 4I), which are located at the enhancers of CD40 and ZAP70 genes, respectively, compared to the CON. MLT treatments were also seen to significantly inhibit the recruitment of active co-factor RNA polymerase II (Figure 4J) and RNA polymerase II serine 5 phosphorylated (Ser5 Pol-II) (Figure 4K) to the target enhancers of CD40 and ZAP70 genes compared to the CON.

## 4. Discussion

The pineal gland and other body organs release MLT, which regulates circadian rhythms and is often used to treat headaches, migraines, jet lag, and insomnia [30,31]. MLT has been established to work well for many conditions, including cancer, liver diseases, and injuries [11,32]. A previous study suggests that MLT is important for immune regulation [33]. It also described the important functions of MLT in vivo intestinal defense [34]. The MLT has no obvious on the cell number and growth compared with the CON. Notably, previous studies on cancer cells have shown that MLT reduces cell viability and is more likely to act as an anticancer drug [35]. Nevertheless, we wanted to address the protective role of normal cell lines to prevent cell death. Indeed, it has been established that MLT plays distinct functions in the growth and death of various cell types [36]. In this case, MLT administration showed no cleavage of caspase 3/7 involved in apoptosis compared to the CON. It cannot be denied that future studies on the dose-dependent effects of MLT need to be conducted. Similarly, the effects of MLT on other cell types are also worth looking forward to and warrant attention.

RNA-seq was used in this work to examine the differential gene expression patterns of treatment groups to further identify potential causes of MLT. The findings demonstrated quantitative variations in DEGs between the comparisons of the two groups. To examine the roles of the genes, KEGG enrichment analysis of the DEGs was carried out. It was found that the DEGs are primarily related to immune and inflammatory responses. It was discovered that the immunological and inflammatory responses are the main factors influencing the DEGs. Both the NF-κB signaling pathway and primary immunodeficiency play important roles in inflammation and the immune response. Regulation of immune response, inflammation, stress response, differentiation, apoptosis, and cell survival is significantly influenced by the NF-κB signaling system [37]. The term primary immunodeficiency describes a variety of diseases caused by abnormalities in the development and/or function of the immune system. They are often classified as either innate immunity (such as phagocyte and complement disorders) or adaptive immunity (such as T cell, B cell, or combined immunodeficiency) [38].

In addition, related primary immunodeficiency and NF-κB signaling pathway genes such as CD40 and ZAP70 significantly increase after MLT treatments in cIECs, which is in accordance with the previous study [23]. CD40 upregulation is significant as it induces proinflammatory responses downstream of this receptor [39]. As CD40-driven ICAM-1 expression and CCL2 production are essential for the pathogenesis of inflammatory diseases, including diabetic retinopathy, advanced glycation end products-mediated CD40 dysregulation is functionally relevant [40]. CD40 ligation increased membrane-bound and soluble CX3CL1 and TNF-α in human aortic and umbilical vein endothelial cells. In contrast, upon CD40 ligation, human retinal endothelial cells did not release TNF-α or upregulate CX3CL1 [39]. Additionally, the interaction between CD40L and its counter-receptor, CD40, activates distinct signaling pathways based on cell type [41,42]. Previous research shows that TNF-α can activate the NF-κB signaling pathway to increase CD40 expression [43,44]. Regarding cytokines that can induce CD40 expression, we can verify previous observations in human-cultured endothelial [45,46], smooth muscle cells [47] and IFNγ, IL-1β, and TNFα, especially in the context of their synergistic effects IFNγ and TNFα induce an increase in CD40 immunoreactivity. Numerous investigations have established that ZAP70 is a major prognostic factor [48,49]. The 70 kDa Zap-70 protein, a member of the Syk family of protein tyrosine kinases, was shown to be an essential component of signaling downstream of the T-cell receptor [50]. Investigation of the functions of the Zap-70 protein in the B-lymphocyte environment revealed its expression and association with a poor prognosis in some subsets of B-lymphocyte-derived malignancies. Several tumor types have significant levels of ZAP70 expression [51]. It promotes B cell receptor signaling, cell proliferation, and migration into the tumor microenvironment [52,53]. According to the literature, ZAP70 stimulates cell migration and invasion of prostate cancer cell lines [54]. In addition, ZAP70 was found to be a prognostic marker for prostate adenocarcinoma [55] and colon cancer response to radiation [56], as well as cervical squamous cell carcinoma [57]. Furthermore, ZAP70 may be an important regulator of metastasis in prostate cancer, as reported by Sun et al. [55].

Histone modification also affects the immune system by changing the expression of some genes. Chemokines, antigen-presenting molecules, and cytokine-encoding genes are examples of such dysregulated genes. Histone modifications regulate gene expression in two ways: they can either promote or decrease immune-responsive genes [58]. Histone modifications can inhibit cytokine and chemokine expression, reducing immune responses [59]. This will lead to a decrease in tissue damage and immunological reactions, allowing the disorders to continue in the body. One significant type of post-transcriptional modification prevalent in organisms is histone methylation. Histone H3 Lys 4 (H3K4), H3K9, H3K27, H3K36, H3K79, and H4K20 are some of the methylations that have been investigated more extensively [60]. H3K4, H3K36, and H3K79 are generally found in chromatin areas of transcriptionally active genes and have activating roles. On the opposing, H3K9, H3K27, and H4K20 are mainly linked with gene expression silencing and often act as repressive markers [61]. The reversible post-translational alteration known as histone acetylation (HAT) includes the transfer of the acetyl moiety of acetyl-CoA to Lys (K) residues. HDACs catalytically reverse this development [62]. In this study, we measured histone acetylated marks by ChIP-qPCR in the MLT-treated cells, and we found that H3K9ac, H3K18ac, and H3K27ac were significantly upregulated at the target loci of ZAP70 and CD40 compared to the CON. In addition, histone methylated marks such as H3K4me1 and H3K4me3 were also upregulated at the target loci of ZAP70 and CD40 in the MLT group compared to the CON, but this change did not involve H3K9bhb and H3K18bhb. H3K9ac and H3K18ac are associated with active transcription of genes that can enhance immune responses. However, their dysregulation may result in the immune-related and tumor-suppressive genes being silenced, aiding in immune evasion [63]. The activation of oncogenes and the inhibition of immune signaling pathways are linked with increased H3K27ac levels, leading to a tumor microenvironment that is less susceptible to immunological responses [64]. Moreover, during T helper cell development, H3K4me1 alterations control gene expression under the effect of TCR signaling [65], thereby influencing immune-related gene expression [66]. Based on the research, precise control over it is essential for healthy development and prevention of disease. H3K4me3 is enriched at transcription start sites and is associated with active gene expression. Upregulated H3K4me3 peaks have been identified in immune response genes in systemic lupus erythematosus (SLE), which correlates with increased expression of immune-related genes such as OAS1 and IFI44L [67]. Both H3K4me1 and H3K4me3 are crucial for immune-related gene expression, and their dysregulation can contribute to immune evasion mechanisms in various diseases.

Among the three RNA polymerase enzymes found in eukaryotic cells, RNA polymerase II is in charge of the transcription of all mRNAs and a significant amount of snRNAs. High efficiency and precision are required for RNA polymerase II transcription to streamline the entire process, from initiation to mRNA co-processing and termination [68]. For polymerase activity, RNA pol II has a carboxy-terminal domain composed of heptapeptide repeats [69]. Serine (Ser) and threonine residues in these repetitions are phosphorylated in RNA polymerase, which is actively transcribing. Moreover, RNA pol II forms the polymerase’s DNA binding domain, a groove where the DNA template is translated into RNA and several other polymerase subunits [70]. Reversibly phosphorylating the CTD of RNA pol II’s major subunit occurs during the transcription cycle. Unphosphorylated RNA pol II is recruited to the promoter, while hyperphosphorylated RNA pol II is involved in active transcription [71]. Within the heptapeptide repeat, phosphorylation takes place at Ser 2, Ser 5, and Ser 7 [72]. The phosphorylation of RNA pol II Ser 5 is specific to promoter regions and required to initiate transcription [73]. Ser5 phosphorylation and RNA polymerase II (Pol II) are crucial for regulating transcription, particularly if the ZAP70 and CD40 signaling pathways are involved. Our finding revealed that the relative enrichment of RNA pol- II and Ser5 pol- II at the target locus of ZAP70 and CD40 was upregulated in the MLT group compared to the CON in the canine ileum epithelial cells. Effective immunological signaling requires the phosphorylation of these proteins, indicating that MLT may improve T cell responses through these processes. While MLT shows promise in enhancing immune function, its effects might differ based on the cell type and the presence of other signaling molecules, indicating a complex interplay in immune modulation.

## 5. Conclusions

As a summary shown in Figure 5, our study provides compelling evidence that MLT treatments exert a significant influence on the regulation of inflammatory genes in canine ileum epithelial cells (cIECs) through histone modification mechanisms. Notably, MLT does not adversely affect cell survival, yet it substantially alters the expression of key genes within inflammatory pathways. Specifically, MLT upregulates the expression of ZAP70 and CD40, which are crucial for immune response, while simultaneously downregulating other genes associated with immune responses. Additionally, MLT enhances the activity of NF-κB signaling pathways, which is instrumental in its anti-inflammatory effects.

The observed increases in histone acetylation and methylation marks at specific gene loci imply that MLT may foster a more favorable transcriptional environment for genes that play a role in inflammation regulation. These findings suggest that MLT could serve as a promising therapeutic agent for managing intestinal inflammatory conditions in dogs. Furthermore, they highlight the necessity for additional in vivo studies to fully elucidate the anti-inflammatory properties of MLT and to explore its potential as a treatment option for inflammatory disorders in veterinary medicine.

## Figures and Tables

**Figure 1 vetsci-12-00087-f001:**
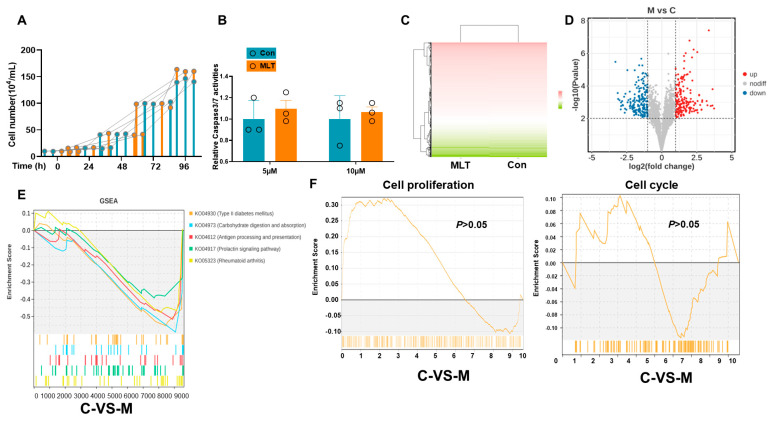
MLT treatment does not affect cell survival and growth in canine epithelial cells. (**A**): MLT treatment did not significantly affect cell growth in canine ileum epithelial cells. (**B**) MLT treatment did not substantially affect cell death detected by caspase3/7 activities in canine ileum epithelial cells. (**C**,**D**) The whole-genome expression heat map and differential gene volcano plot of the CON and MLT group reflect the overall impact of MLT treatment on gene expression; the results show no significant differences between the two groups. (**E**) GSEA dynamic analysis to draw the top ranked pathways with significant differences in ES score. (**F**) There was no significant upregulation or downregulation of the cell cycle and proliferation genes due to MLT treatment in the cells (*p* > 0.05).

**Figure 2 vetsci-12-00087-f002:**
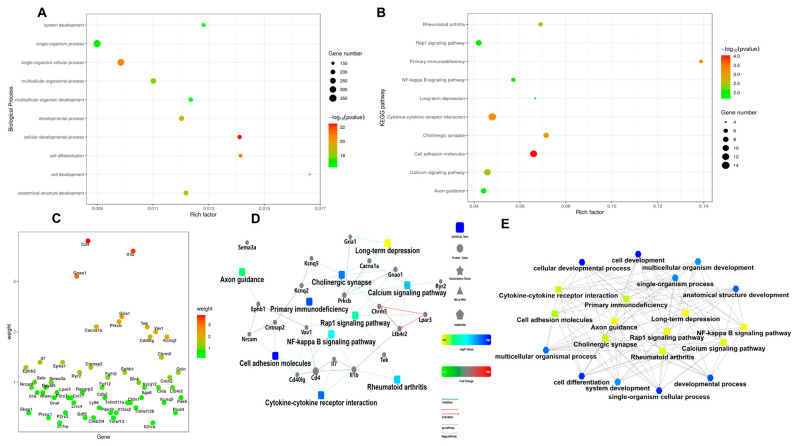
**Functional annotation pathway enrichment result.** (**A**–**C**) To “Enrichment of biological processes and pathways for differential genes, focusing on significant enrichment of primary immunodeficiency and NF-κB signaling pathways. (**D**) PPI (protein interaction) network diagram of significantly changed pathways and related genes enriched by KEGG. (**E**) Network diagram of the interaction between enriched pathways and cellular processes. Panels (**D**,**E**) highlight genes, pathways, and processes for further exploration.

**Figure 3 vetsci-12-00087-f003:**
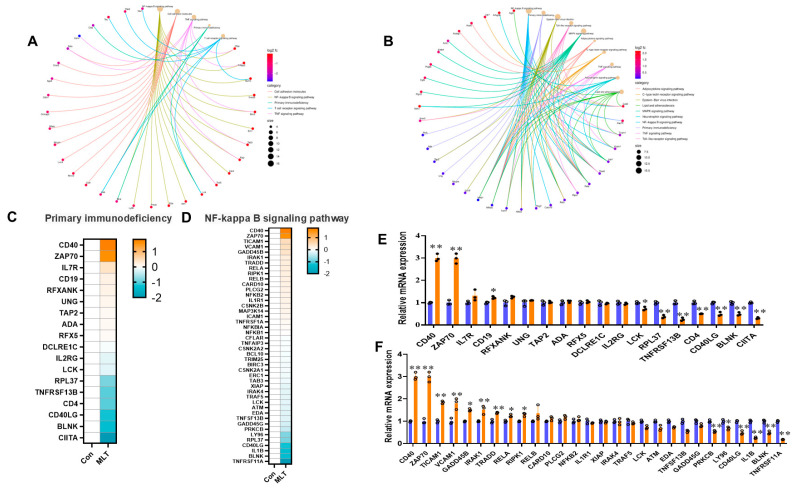
**The gene expression of primary immunodeficiency and NF-κB signaling pathway genes of cIECs affected by MLT treatment**. Gene expression of primary immunodeficiency and NF-kappa B signaling pathway genes in cIECs affected by MLT treatment. (**A**) Functional network diagram of upregulated differential genes in inflammation- and immune-related pathways. (**B**) Functional network diagram of downregulated differential genes in inflammation- and immune-related pathways. (**C**) Heat map of core gene expression in primary immunodeficiency pathways. (**D**) Heat map of core gene expression in NF-κB signaling pathways. (**E**,**F**) Relative mRNA expression of inflammation- and immune-related pathway genes, consistent with sequencing results. Data are presented as means ± SEM (n = 3). * *p* < 0.05; ** *p* < 0.001.

**Figure 4 vetsci-12-00087-f004:**
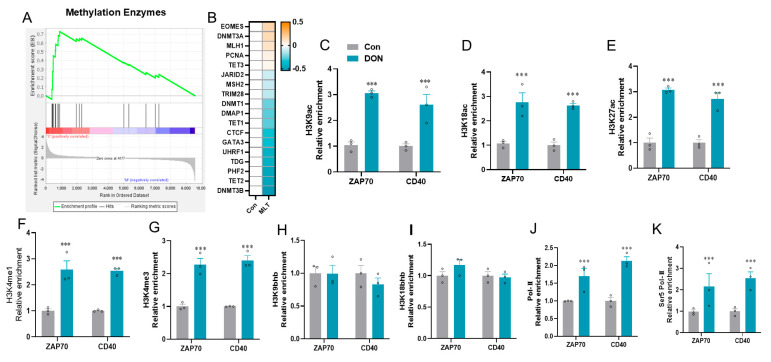
**Histone modifications on the ZAP70 and CD40 genes.** (**A**) Demethylation pathway enrichment in MLT-treated cells using GSEA. (**B**) Heat map of core gene expression of demethylation enzymes. (**C**–**E**) Histone acetylated marks measured by ChIP-qPCR in the MLT-treated cells. (**F**,**G**) Histone methylated marks measured by ChIP-qPCR. (**H**,**I**) Histone hydroxybutyrylated marks measured by ChIP-qPCR. (**J**,**K**) RNA polymerase II and phosphorylation at Ser5 measured by ChIP-qPCR. Data are indicated as means ± SEM (n = 3). *** *p* < 0.0001.

**Figure 5 vetsci-12-00087-f005:**
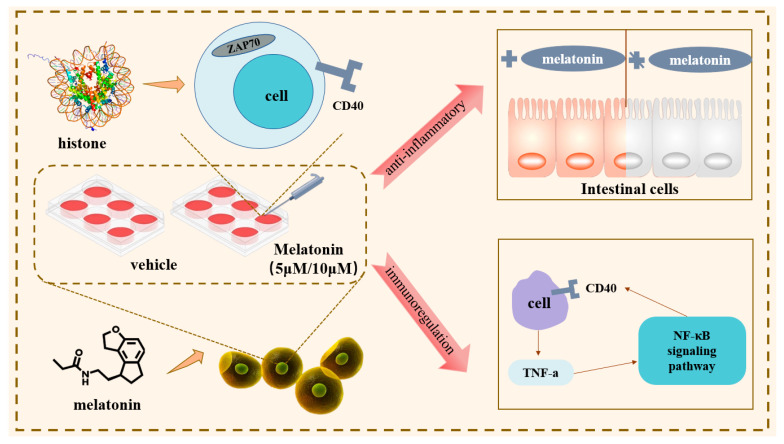
Melatonin treatment could upregulate the expression of ZAP70 and CD40 and other core genes related to inflammation and immune response via histone modification.

## Data Availability

The original transcriptome data proposed in this study has been preserved at the National Center for Biotechnology Information (https://www.ncbi.nlm.nih.gov/, accessed on 26 September 2024), and the preservation number is PRJNA1165463.

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
