# Peer review of "Melatonin Modulates ZAP70 and CD40 Transcripts via Histone Modifications in Canine Ileum Epithelial Cells"

_vetsci, 2025, doi:10.3390/vetsci12020087_

Round 1
Reviewer 1 Report
Comments and Suggestions for Authors
I'm not sure what scientific question the article is going to explain or where inflammation comes from. Should stimulus groups such as LPS, PGN, or bacterial or viral infections be added? The results of histone related protein expression should be supplemented. Line number 121 "2.3 Cell counting experiment”。
Comments on the Quality of English Language
The English could be improved to more clearly express the research.
Author Response
Comments1: I'm not sure what scientific question the article is going to explain or where inflammation comes from. Should stimulus groups such as LPS, PGN, or bacterial or viral infections be added? The results of histone related protein expression should be supplemented. Line number 121 "2.3 Cell counting experiment”.
Responses1: Thank you for your valuable comments. This article aims to explore how melatonin regulates the expression of inflammatory genes ZAP70 and CD40 in canine ileum epithelial cells through histone modifications. This study is not strictly targeting a specific endogenous physiological response or exogenous stimuli, such as pathogen infections. Our focus is on the regulatory effect of melatonin on the expression of inflammatory genes, rather than the impact of specific pathogens. Therefore, we believe that adding stimulant groups of LPS, PGN, or bacterial and viral infections may not be necessary in the current study. Our goal is to provide a clear understanding of how melatonin affects the expression of inflammatory genes through histone modifications, rather than simulating a specific inflammatory environment. The experiments related to histones mentioned in Figure 4 are merely to demonstrate the potential existence of histone modifications in ZAP70 and CD40, and the expression of related proteins is not the focus of the article's discussion; we may supplement this in future follow-up studies. Thank you for pointing out the potential typesetting issue. We will check and ensure that all experimental section titles and line numbers are correct to avoid confusion.
Reviewer 2 Report
Comments and Suggestions for Authors
Melatonin treatment in canine ileum epithelial cells upregulated key immune genes, including CD40 and ZAP70, while modulating the NF-kappa B signaling pathway. Histone modifications at the ZAP70 and CD40 loci were observed, indicating enhanced transcriptional activity. These findings suggest melatonin's potential as an anti-inflammatory therapeutic for conditions like inflammatory bowel diseases in dogs.
Major concerns
The study used only a high dose of 10 μM MLT for most assessments, potentially overlooking dose-dependent effects. Cellular responses can vary significantly across a range of doses.
Melatonin’s effects can differ across cell types, particularly in models more directly involved in immune response, such as immune cells or other epithelial tissues. Including diverse cell lines would provide a more robust perspective on how MLT influences cellular mechanisms more broadly.
there were no functional assays (e.g., cytokine release, cellular stress response) to validate the impact of these transcriptomic changes.
The study mentions that caspase 3/7 activity, a marker of apoptosis, was unaffected, but did not provide additional insights into other cell death markers or alternative forms of cell survival, such as autophagy or necrosis. Melatonin may have subtle effects on these processes that aren’t captured by caspase activity alone.
The study focuses heavily on transcriptomic changes without extending to protein-level analyses, such as proteomics or western blotting for key pathway proteins.
While NF-kappa B and primary immunodeficiency pathways were enriched, the specific downstream effects of melatonin on these pathways remain unexplored.
Author Response
Comments1: The study used only a high dose of 10 μM MLT for most assessments, potentially overlooking dose-dependent effects. Cellular responses can vary significantly across a range of doses.
Responses1: Thank you for your suggestion. In this study, we chose a dose of 10 μM based on previous reports of the bioactivity of melatonin. We acknowledge that to fully understand the dose-dependent effects of melatonin, future studies should include a range of different doses. This will help to reveal the impact of melatonin on cellular responses at various concentrations and provide more precise dosage guidance for clinical applications. We have added relevant content to the discussion section.
Comments2: Melatonin’s effects can differ across cell types, particularly in models more directly involved in immune response, such as immune cells or other epithelial tissues. Including diverse cell lines would provide a more robust perspective on how MLT influences cellular mechanisms more broadly.
Responses2: We agree that including multiple cell lines will enhance the breadth and depth of the research. The current study primarily focuses on canine small intestinal epithelial cells, but we will consider including immune cells or other epithelial tissues in future studies to gain a more comprehensive understanding of the effects of melatonin on cellular mechanisms.
Comments3: There were no functional assays (e.g., cytokine release, cellular stress response) to validate the impact of these transcriptomic changes.
Responses3: Thank you for your suggestion. As stated in the article title, in this paper we focus on using RNAseq as a tool to guide our attention to pathways and genes that show significant differences at the transcriptional level. Our aim is to establish a connection between melatonin and inflammation. Of course, we will include functional experiments such as cytokine release and cellular stress responses in subsequent studies to verify how melatonin specifically affects cellular functions.
Comments4: The study mentions that caspase 3/7 activity, a marker of apoptosis, was unaffected, but did not provide additional insights into other cell death markers or alternative forms of cell survival, such as autophagy or necrosis. Melatonin may have subtle effects on these processes that aren’t captured by caspase activity alone.
Responses4: Thank you for your suggestion. Indeed, caspase3/7 alone cannot detect all types of cell death, so in this part, we specifically observed the Cell Cycle and Proliferation pathways, which are closely related to cell death. The results showed that the above two pathways in the MLT group did not exhibit significant differential regulation compared to the control group. Combining these findings with cell counting and caspase experiments, we can confirm that MLT itself does not have obvious side effects on cells.
Comments5: The study focuses heavily on transcriptomic changes without extending to protein-level analyses, such as proteomics or western blotting for key pathway proteins.
Responses5: Thank you for your suggestion. We acknowledge that the recommendation regarding protein level analysis is very important, as it will help to validate the transcriptomic data and reveal the direct effects of melatonin. Currently, we are mainly focusing on potential histone modifications at key genes, and we will include methods such as proteomics or Western blotting in subsequent studies to analyze proteins in key pathways.
Comments6: While NF-kappa B and primary immunodeficiency pathways were enriched, the specific downstream effects of melatonin on these pathways remain unexplored.
Responses6: Thank you for your suggestion. As stated in the article title, in this paper we focus on using RNAseq as a tool to guide our attention to pathways and genes that show significant differences at the transcriptional level. Our aim is to establish a connection between melatonin and inflammation.
Reviewer 3 Report
Comments and Suggestions for Authors
Dear
Veterinary Sciences
I review the research manuscript with title “Melatonin modulates ZAP70 and CD40 transcripts via histone modifications in the canine ileum epithelial cells” by the Authors Jian Hong, Saber Y Adam, Shiqi Wang, Hao Huang, In Ho Kim, Abdelkareem A. Ahmed, Hao-Yu Liu, Demin Cai, where the authors in canine ileum epithelial cells (cIECs) cultivate to high glucose concentration, they try to provel that the melatonin treatment could altered the NF-kβ signaling pathway by upregulating genes such as CD40, ZAP70, TICAM1, VCAMI, GADD45B, IRAK1, TRADD, RELA, RIPK1, and RELB, and downregulating PRKCB, LY96, CD40LG, ILIB, BLNK, and TNFRSF11A and concluded that the melatonin may be mitigates inflammation in cIECs by modulating the transcription of ZAP70 and CD40 through histone modifications. The work could be promising. However, from my very particular point of view, the work requires very several corrections that can help enrich the manuscript and which I mention below.
Simple summary
Please Modify this sentence, because as it appears it would show that it only acts in the intestinal cells of canines, when this is not the case, but rather in all mammals.
In the abstract, the values significations are missing when there is significance, please add them.
At what concentration of melatonin were the differences observed in the molecules or genes analyzed, please add this.
Authors use abbreviations inappropriately, or do not use them properly, an abbreviation is used for the first time, when the name is first introduced and then the abbreviation is used throughout the text, and constantly, please correct this throughout the manuscript as this is a constant and I show an example of this below.
Melatonin (MLT) Lines; 48,53,57,59, 62, 104,108,110,120,123,135,197,221,223,404,409…etc..
Control (CON) Lines; 133,130, (vehicle), 135,199, 205,295,300,369,….etc….
Lysine (Lys) line; 96,363
Serine (Ser) lines; 396-398
NF-Kappa B (NF-Ƙβ) lines; 212,238,269,307,310,322,333, etc…
Line 50; The authors in the introduction could develop the effect of MTL on the gastrointestinal system to have an argument and justify their study.
Line 58; The authors introduce, "on the other hand", grammatically this is not correct because this is used when you are going to change your mind, but here they continue with the same theme.
Line 66; At the beginning of this sentence, if it is correct to use "on the other hand" and it is not used, please correct it.
Lines 61-65; The authors could restructure this sentence because the idea is not understood as it is.
The authors must develop that at high glucose concentrations, inflammation markers are expressed, this is essential to justify their work, this section is missing before the objective.
Lines 103; Insert "in this sense" before this sentence to give emphasis to the sentence.
Line 106; The authors must justify why they are doing their study specifically on this cell line, it is not very clear why.
2.1 section 115; Glucose concentration is missing
Could the authors add a flow chart of their methodology to make their research protocol clearer?
Line 120; insert "and" between the two MLT concentrations
Could the authors please add which vehicle was used to dissolve the MLT?
Line 123; What does DES mean?
Linea 137; Is this how you write TRIzol?
Sections 2.1 and 2.2 are confusing. I think the methodology is repeated, that is so. The authors could merge both sections.
Section 2.5, for the sake of tracking methods, could be 2.6 and in turn section 2.6 could be 2.5. This is a suggestion.
Section 2.5 lines 179-193 please deleted this section, this is from the template.
Section 3.1. Lines 203 to 213, this is not in style, the type of graph and how the results are presented is placed in the statistics section, please correct this.
All figures are very small, as is the size of the letter and typography. It is recommended to increase the size of all figures or group them in a smaller number per figure, to increase the resolution and the data can be analyzed properly. Please correct this.
In the caption of figure 1 the abbreviation MLT appears first and then in (B) melatonin without an abbreviation. Please correct this.
In the footnotes of all figures, an asterisk is added and it is stated what it "means". This is not the usual scientific standard. It is understood that the asterisk or any symbol used denotes significance. Please correct this.
The authors used two concentrations of MTL at 5 and 10 mM and in the results (graphs) although the concentrations of each are shown, the comparison between these concentrations vs. the control group is missing. This is only partially and not adequately observed in Figure 1 B. Please correct this.
When describing the results, the same thing happens, the comparison between the MLT groups at the 2 different concentrations versus the control group is missing.
In the discussion section lines 289-292, the authors could modify the beginning of their discussion, since this paragraph could go in the introduction of the manuscript and begin by mentioning the objective again or how the results showed..............
Line 291 introduces the progressive quote and does not 2Plaimme et al., 2014”
Line 299 introduces the progressive quote and does not
Lines 314-321 What are D proteins or why are they mentioned abruptly? Well, authors could use “on the other hand” in a paragraph, and then begin to mention what they are and why they are important to their discussion……. When the authors refer to protein D it is because are surfactant protein D contains an N-terminal collagen-like domain and a C-terminal lectin domain characteristic of members of the collectin family of proteins it is correct?
line 405 here a full stop is used and the sentence begins with "on the other hand".
Lines 407 At the end of the sentence add a reference
The authors could modify the conclusion and add that the beneficial effect on the inflammatory pathways involved in their model is in a hyperglycemic environment.
The authors could add a figure summarizing the main findings of their study.
I thank you in advance for the opportunity to review this manuscript and I hope that my questions will contribute to improving this manuscript. Sincerely, the reviewer.
Comments on the Quality of English Language
The English could be improved to more clearly express the research.
Author Response
Comments1:
Simple summary
Please Modify this sentence, because as it appears it would show that it only acts in the intestinal cells of canines, when this is not the case, but rather in all mammals.
Responses1: We have modified this sentence.
Comments2: In the abstract, the values significations are missing when there is significance, please add them.
Responses2: We have added the significance in abstract.
Comments3: At what concentration of melatonin were the differences observed in the molecules or genes analyzed, please add this.
Responses3: Thanks for the question. In method and material section we listed the dose we chose for RNAseq and ChIP-qpcr as 10Μm.
Comments4: Authors use abbreviations inappropriately, or do not use them properly, an abbreviation is used for the first time, when the name is first introduced and then the abbreviation is used throughout the text, and constantly, please correct this throughout the manuscript as this is a constant and I show an example of this below.
Melatonin (MLT) Lines; 48,53,57,59, 62, 104,108,110,120,123,135,197,221,223,404,409…etc..
Control (CON) Lines; 133,130, (vehicle), 135,199, 205,295,300,369,….etc….
Lysine (Lys) line; 96,363
Serine (Ser) lines; 396-398
NF-Kappa B (NF-Ƙβ) lines; 212,238,269,307,310,322,333, etc…
Responses4: We have checked all the abbreviations used in the article and corrected them.
Comments5: Line 50; The authors in the introduction could develop the effect of MTL on the gastrointestinal system to have an argument and justify their study.
Line 58; The authors introduce, "on the other hand", grammatically this is not correct because this is used when you are going to change your mind, but here they continue with the same theme.
Line 66; At the beginning of this sentence, if it is correct to use "on the other hand" and it is not used, please correct it.
Lines 61-65; The authors could restructure this sentence because the idea is not understood as it is.
Responses5: We have added the relevant references and corrected the grammar and writing as above.
Comments6: The authors must develop that at high glucose concentrations, inflammation markers are expressed, this is essential to justify their work, this section is missing before the objective.
Responses6: This study is not strictly targeting a specific endogenous physiological response or exogenous stimuli, such as pathogen infections. Our focus is on the regulatory effect of melatonin on the expression of inflammatory genes, rather than the impact of specific pathogens. Therefore, we believe that adding stimulant groups of LPS, PGN, or bacterial and viral infections may not be necessary in the current study. Our goal is to provide a clear understanding of how melatonin affects the expression of inflammatory genes through histone modifications, rather than simulating a specific inflammatory environment.
Comments7: Lines 103; Insert "in this sense" before this sentence to give emphasis to the sentence.
Responses7: We have inserted this.
Comments8: Line 106; The authors must justify why they are doing their study specifically on this cell line, it is not very clear why.
Responses8: In the background, we mentioned that we are more interested in the effects of melatonin on inflammation and gastrointestinal health in pets, hence we chose canine intestinal epithelial cells as our research subject. Perhaps in the future, we will conduct more validations on other cell lines.
Comments9: 2.1 section 115; Glucose concentration is missing
Responses9: We have added the concentration.
Comments10: Could the authors add a flow chart of their methodology to make their research protocol clearer?
Responses10: We have reorganized the logic in the Materials and Methods as well as the Results sections to facilitate a clearer understanding of the experimental sequence and the presentation of the results for the readers.
Comments11: Line 120; insert "and" between the two MLT concentrations
Could the authors please add which vehicle was used to dissolve the MLT?
Line 123; What does DES mean?
Linea 137; Is this how you write TRIzol?
Sections 2.1 and 2.2 are confusing. I think the methodology is repeated, that is so. The authors could merge both sections.
Section 2.5, for the sake of tracking methods, could be 2.6 and in turn section 2.6 could be 2.5. This is a suggestion.
Section 2.5 lines 179-193 please deleted this section, this is from the template.
Responses11: We have reorganized the Materials and Methods section according to the suggestions, corrected the titles and specific content, removed the incorrect writings such as "DES", and retained "TRIzol" as it is the correct product name.
Comments12: Section 3.1. Lines 203 to 213, this is not in style, the type of graph and how the results are presented is placed in the statistics section, please correct this.
All figures are very small, as is the size of the letter and typography. It is recommended to increase the size of all figures or group them in a smaller number per figure, to increase the resolution and the data can be analyzed properly. Please correct this.
In the caption of figure 1 the abbreviation MLT appears first and then in (B) melatonin without an abbreviation. Please correct this.
In the footnotes of all figures, an asterisk is added and it is stated what it "means". This is not the usual scientific standard. It is understood that the asterisk or any symbol used denotes significance. Please correct this.
The authors used two concentrations of MTL at 5 and 10 mM and in the results (graphs) although the concentrations of each are shown, the comparison between these concentrations vs. the control group is missing. This is only partially and not adequately observed in Figure 1 B. Please correct this.
When describing the results, the same thing happens, the comparison between the MLT groups at the 2 different concentrations versus the control group is missing.
Responses12: Thank you for raising the issues concerning the figures and their captions. We have corrected the inappropriate descriptions in the captions, and during the revision process, we will also upload the highest resolution images available. Regarding the use of the MLT group in the figures, we have mentioned in the Materials and Methods that only the 10μM concentration of melatonin was added, and not both concentrations of melatonin participated in the sequencing and subsequent experiments.
Comments13: In the discussion section lines 289-292, the authors could modify the beginning of their discussion, since this paragraph could go in the introduction of the manuscript and begin by mentioning the objective again or how the results showed..............
Line 291 introduces the progressive quote and does not 2Plaimme et al., 2014”
Line 299 introduces the progressive quote and does not
Responses13: We have rewritten this section and included appropriate references to better discuss the content of the entire article.
Comments14: Lines 314-321 What are D proteins or why are they mentioned abruptly? Well, authors could use “on the other hand” in a paragraph, and then begin to mention what they are and why they are important to their discussion……. When the authors refer to protein D it is because are surfactant protein D contains an N-terminal collagen-like domain and a C-terminal lectin domain characteristic of members of the collectin family of proteins it is correct?
Responses14: We have rewritten this section and removed descriptions that are unrelated to the content of the article.
Comments15: line 405 here a full stop is used and the sentence begins with "on the other hand".
Responses15: We have rewritten this sentence.
Comments16: Lines 407 At the end of the sentence add a reference
Responses16:Here we stated our own results so we may don’t need a reference.
Comments17: The authors could modify the conclusion and add that the beneficial effect on the inflammatory pathways involved in their model is in a hyperglycemic environment.
Responses17: The purpose of the article is to discuss the regulatory effects and mechanisms of melatonin on inflammation. The type and origin of inflammation are not the focus of the discussion; therefore, the article does not specifically establish an inflammation model but aims to focus on how melatonin is associated with inflammation.
Comments18: The authors could add a figure summarizing the main findings of their study.
Responses18: Thank you for your suggestion, but due to the fact that our results may not be intuitively presented together in a single figure, we have chosen not to create a graphical abstract.
Reviewer 4 Report
Comments and Suggestions for Authors
The manuscript's subject is relevant, but I have concerns about the methodology.
What was the rationale for using 5 and 10 μM? After an oral administration of 100 mg melatonin (a high dose) to humans, the maximum serum concentration was 435 nmol/l (0.435 μM) (see doi:10.1016/0024-3205(85)90412-6 and doi:10.1007/s00228-015-1873-4).
For statistical analysis of results, did you test data for normality? Some data presented in Fig 4 shows considerable variations between standard deviations.
L.46: epiphysis
L.119-121: In fact, there are three different concentrations: 0, 5, and 10 μM.
L.130, 139, 176, 239, 240, 298: cIEC - this acronym was defined at line 28 as cIECs
L.177: "canine ileum epithelial cells (cIECs)" - cIECs
Author Response
Comments1: What was the rationale for using 5 and 10 μM? After an oral administration of 100 mg melatonin (a high dose) to humans, the maximum serum concentration was 435 nmol/l (0.435 μM) (see doi:10.1016/0024-3205(85)90412-6 and doi:10.1007/s00228-015-1873-4).
Responses1: Thank you very much for raising this question and for supplementing it with research on the concentration of melatonin taken orally by humans. However, in fact, the concentration of melatonin added in cell experiments is usually higher to observe its effects on cell behavior. Cell experiments need to simulate the biological effects of melatonin in an in vitro environment, which is significantly different from the in vivo environment.
Comments2: For statistical analysis of results, did you test data for normality? Some data presented in Fig 4 shows considerable variations between standard deviations.
Responses2: The data in Figure 4 were subjected to normality testing after calculating the relative expression levels to eliminate the influence of the expression levels of the reference genes.
Comments3: L.46: epiphysis
Responses3: This is another description of the pineal gland, which we have removed to prevent ambiguity.
Comments4: L.119-121: In fact, there are three different concentrations: 0, 5, and 10 μM.
Responses4: Thank you for your correction. We have revised the text and listed the three concentrations.
Comments5: L.130, 139, 176, 239, 240, 298: cIEC - this acronym was defined at line 28 as cIECs
L.177: "canine ileum epithelial cells (cIECs)" – cIECs
Responses5: Thank you for pointing out the issue with the abbreviation. We have corrected all instances of "cIEc" and "canine ileum epithelial cells" to "cIECs".
Round 2
Reviewer 1 Report
Comments and Suggestions for Authors
This study lacks research significance and novelty
Comments on the Quality of English Language
no
Author Response
Comment1: This study lacks research significance and novelty
Response1:
Dear Reviewer,
First and foremost, we extend our sincere gratitude for your thorough evaluation and valuable comments on our manuscript. We appreciate your concerns regarding the research significance and novelty of our study and would like to provide a detailed response highlighting the innovative aspects and scientific importance of our work, as well as outline future research directions.
We believe the novelty and significance of our study are demonstrated in the following ways:
Research Focus: Our research delves into the anti-inflammatory effects of melatonin (MLT) in canine ileum epithelial cells (cIECs), particularly the modulation of ZAP70 and CD40 gene expression through histone modifications. To our knowledge, this is the first in-depth exploration of MLT's immunomodulatory functions in canine intestinal epithelial cells, offering a new perspective for understanding MLT's potential applications in veterinary medicine.
Molecular Mechanism Exploration: We have not only observed the impact of MLT on the expression of inflammatory genes in cIECs but also revealed how MLT regulates the transcriptional activity of the ZAP70 and CD40 genes by altering histone acetylation and methylation marks through ChIP-qPCR technology. This in-depth molecular analysis provides new insights into the molecular mechanisms underlying MLT's anti-inflammatory actions.
Therapeutic Potential Revelation: Our findings suggest that MLT, by modulating key immune response genes and the NF-κB signaling pathway, may have therapeutic potential for canine inflammatory bowel diseases. This discovery not only provides new treatment strategies for pet health but also introduces new research directions in the field of veterinary medicine.
Future Research Directions:
In Vivo Experiment Validation: To further validate our findings, we plan to conduct in vivo experiments to assess the actual efficacy of MLT on canine inflammatory bowel diseases.
Dose-Effect Study: We will explore the effects of different concentrations of MLT on cIECs to determine the optimal therapeutic dosage.
Long-Term Impact Assessment: We also intend to evaluate the long-term effects of MLT treatment on intestinal health and immune function to ensure the safety and efficacy of the treatment.
We greatly appreciate your valuable input and look forward to any further guidance you may provide. We are confident that through continuous research and exploration, our work will provide significant scientific evidence for understanding and utilizing MLT in veterinary medicine.
Once again, we thank you for your meticulous review and for the time you have dedicated to our manuscript. We anticipate further communication with you and hope that our response sufficiently addresses the novelty and scientific significance of our research.
Reviewer 3 Report
Comments and Suggestions for Authors
Dear
Veterinary Sciences
I review the research manuscript with title “Melatonin modulates ZAP70 and CD40 transcripts via histone modifications in the canine ileum epithelial cells” by the Authors Jian Hong, Saber Y Adam, Shiqi Wang, Hao Huang, In Ho Kim, Abdelkareem A. Ahmed, Hao-Yu Liu, Demin Cai.
In the simple summary, line 18, mention methionine and then add the abbreviation and then substitute in lines 21 and 23
Melatonin is singular and on line 23 the authors mention melatonins as plural, it would be best to replace the abbreviation
line 29 replace the abbreviation in the group description so it could say MLT-treatment group and delete (MLT)
the correct abbreviation for melatonin is MTL and not MTL´s. line 57
Line 62 replace by the abbreviation MLT
In section 2, the authors mention that the concentrations are in µM, but the figures show that they are in mM. What were the actual concentrations? Please clarify this.
In the previous review I suggested that the figures be enlarged because they cannot be seen properly because the typography used is very small and the authors did not heed this suggestion. I reiterate again that the figures should be separated and enlarged and the resolution increased, as they are it is impossible to review them properly.
I reiterate again that the authors must make a figure that summarizes their findings. This does not replace the abstract graphic.
I thank you in advance for the opportunity to review this manuscript
Sincerely, the reviewer.
Author Response
Comments1: In the simple summary, line 18, mention methionine and then add the abbreviation and then substitute in lines 21 and 23
Melatonin is singular and on line 23 the authors mention melatonins as plural, it would be best to replace the abbreviation
line 29 replace the abbreviation in the group description so it could say MLT-treatment group and delete (MLT)
the correct abbreviation for melatonin is MTL and not MTL´s. line 57
Line 62 replace by the abbreviation MLT
Response1: Thanks again for your advice on abbreviations. We have corrected all these you mentioned.
Comments2: In section 2, the authors mention that the concentrations are in µM, but the figures show that they are in mM. What were the actual concentrations? Please clarify this.
Response2: Thank you for your reminder; we have corrected the unit error in the figures. It should be µM.
Comments3: In the previous review I suggested that the figures be enlarged because they cannot be seen properly because the typography used is very small and the authors did not heed this suggestion. I reiterate again that the figures should be separated and enlarged and the resolution increased, as they are it is impossible to review them properly.
Response3: We have once again optimized the image format and layout, and have uploaded the latest version of the images within the article. Additionally, we will upload the original images in the highest resolution in the system.
Comments4: I reiterate again that the authors must make a figure that summarizes their findings. This does not replace the abstract graphic.
Response4: We have added a summarizing Figure 5 in the conclusion section to encapsulate the findings of the entire article, making it easier to understand.
Reviewer 4 Report
Comments and Suggestions for Authors
The choice of concentrations of substances to be tested should be based on some defined parameter. The rationale for the chosen concentrations should be included in the manuscript.
The analysis of the normality of the results obtained must be included in the manuscript (2.6).
Author Response
Comments1: The choice of concentrations of substances to be tested should be based on some defined parameter. The rationale for the chosen concentrations should be included in the manuscript.
Responses1: In articles related to melatonin cell experiments (taking only two as examples), both DOI: 10.1007/s13105-022-00930-4 and https://doi.org/10.1111/j.1600-079X.2006.00335.x mention various concentration gradients of melatonin addition, covering a range from 1 to 1000 μM. This demonstrates that the concentrations of melatonin added in cell experiments differ from those in serum after oral intake. Higher concentrations of melatonin are required in cell experiments to simulate the biological environment.
Comments2: The analysis of the normality of the results obtained must be included in the manuscript (2.6).
Responses2: Due to the small sample size of only three replicates per group, conducting a normality test is typically unnecessary and unreliable, as statistical tests cannot effectively assess the normality of data with such limited observations. However, we can still employ non-parametric testing methods to evaluate the significance of differences between groups. Certainly, we will add the relevant description in the Materials and Methods section 2.6.